# UNAST: Unified framework for Neural Architecture Search for Transformers

## Abstract

We present UNAST, a novel method for optimizing Large Language Models (LLMs) after training. UNAST integrates Neural Architecture Search (NAS) with sparsity and quantization techniques to compress LLMs. Starting from a pre-trained model, it replaces layers (such as attention and MLP) with more efficient alternatives by modifying attention heads, key-value projection dimensions, and MLP expansion ratios. Layer candidates undergo local distillation to replicate the original layers. Scores and costs (e.g., latency, parameter count) for each operator are input into an Integer Linear Optimizer, which determines the optimal architecture under given constraints. Our experiments show that UNAST scales effectively to large models, reducing training costs by up to 10x compared to training smaller models from scratch. Testing on GPT-3 and LLaMa reveals that UNAST enhances latency and memory efficiency by up to 60% with minimal accuracy loss. Additionally, it offers insights into the impact of various compression techniques on Transformer layers, facilitating the creation of non-uniform models.

## 1 Introduction

Natural Language Processing (NLP) has undergone a major transformation with the rise of Large Language Models (LLMs), largely due to the scalability of the attention mechanism. However, deploying these increasingly large models efficiently presents growing challenges. LLM families, like LLaMa with its 7B, 13B, and 70B variants, typically feature multiple versions to accommodate different hardware and time constraints. These models, however, are trained independently, creating redundancy. A more efficient strategy would be to compress larger models down to smaller sizes rather than training each from scratch.

Building on this idea, post-training techniques like quantization and sparsity have emerged to enhance LLM deployment. While they reduce model size, they don't always improve throughput and often require manual hyperparameter tuning. Neural Architecture Search (NAS) can optimize model architectures automatically for efficiency but is resource-intensive when applied to LLMs. Therefore, a comprehensive post-training optimization method is needed.

To tackle this challenge, we introduce UNAST, an optimization framework that leverages NAS to compress models according to specific constraints like latency and parameter count. UNAST adjusts internal layer parameters, including the number of attention heads, Key-Value (KV) projection dimensions, and expansion factors, to achieve efficient compression.

We begin with a pre-trained model and distill teacher layers (attention and MLP) into candidate configurations by adjusting parameters like the number of heads, MLP projections, and attention types. Each candidate is scored based on its similarity to the original teacher blocks. Using integer linear programming (ILP), we select K architectures that optimize for constraints using candidate scores and associated costs. These architectures undergo quick fine-tuning and evaluation, with the top-performing one chosen for a longer final fine-tuning iteration.

We demonstrate several key advantages of our approach:

1. **Time to Create a New Model.** Our method produces new models up to 10 times faster than pre-training from scratch, thanks to efficient layer-wise knowledge distillation and automated candidate model selection.

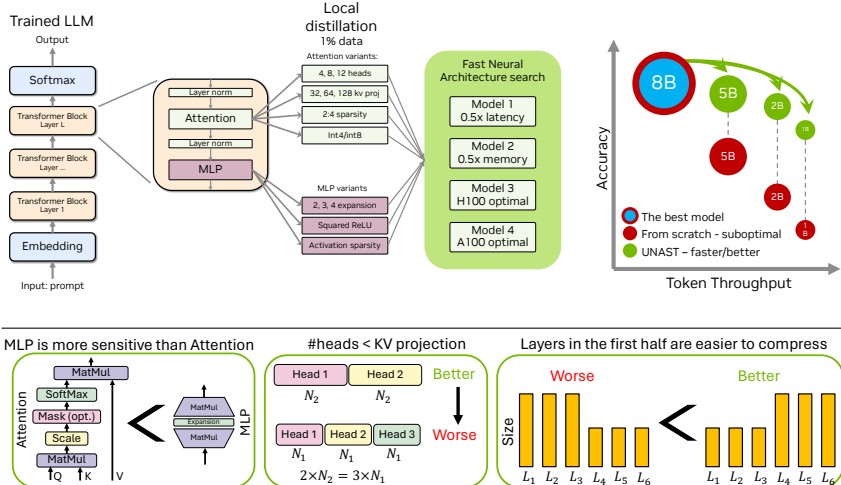

Figure 1: Overview of UNAST as a unified approach to post-training LLM optimization. Starting from a pre-trained model, layers are replaced with more efficient counterparts using a fast NAS approach to find models under user-defined constraints like latency and number of parameters. This results in better, smaller models with a 10x reduction in costs compared to training from scratch. At the bottom, we share the main findings for future architecture design.

2. **Latency and Memory Footprint Improvements**. Models generated by our approach greatly improve the original model's latency and memory footprint. Crucially, target parameters like latency or memory can be customized and set as constraints in the NAS process. These efficiency gains are achieved with minimal accuracy loss, ensuring the models remain highly effective for their tasks.

3. **Detection of Heterogeneity in Model Structure**. Our approach enables heterogeneity in the teacher model's structure, resulting in models with non-uniform blocks. This optimization creates more tailored and efficient architectures that capitalize on the strengths of various model components.

Our main contributions can be summarized as:

1. **Novel Neural Architecture Search (NAS) Approach Applied to Large Language Models**. We present a NAS methodology specifically designed to optimize Transformer-based large language models. This approach refines model architectures, enhancing both performance and efficiency. It effectively addresses the unique challenges posed by the structures of large language models like GPT and LLaMA.

2. **Empirical Insights into Transformer Model Structure**. Through extensive empirical observations, we offer insights into the structure of Transformer models, including analyses of how compression impacts various layers of these architectures.

## 2 RELATED WORK

Compression of neural networks remains an active area of research, as highlighted in recent surveys Cheng et al. (2018); Gou et al. (2020); Park et al. (2024). Common methods include weight sparsification Han et al. (2016); LeCun et al. (1990); Hassibi & Stork (1993), structured pruning via channel removal Molchanov et al. (2017), and quantization Frantar et al. (2022); Lin et al. (2023); Xiao et al. (2022). We focus on LLM compression for faster inference and will review some recent work in this section.

**Sparsity** in the unstructured form is not directly beneficial for GPU inference. SparseGPT Frantar & Alistarh (2023) and Wanda Sun et al. (2023) additionally explore 2:4 structured sparsity that can benefit faster inference. **Structured pruning** is another popular technique where entire heads and MLP channels are removed from the transformer Xia et al. (2022); Ma et al. (2023b); Zhang et al. (2023); Xia et al. (2023); Kurtic et al. (2023a). Such pruning results in a smaller and faster model; however, the entire process needs to be repeated for every target budget. **Depth pruning** removes entire blocks from the model as the residual skip connections will still allow signal propagation. Stochastic depth during training was applied in Fan et al. (2020). Importance-based layer dropping

was studied in Men et al. (2024); Kim et al. (2024); Yuan et al. (2024), and layer merging was explored in Yang et al. (2024). **Quantization** is a common technique to reduce the bitwidth of weight tensors. Round-to-nearest approaches Dettmers et al.; Yao et al. (2022) perform straight-forward per-element quantization, while carefully selecting quantization granularity. AdaRoundNagel et al. (2020) computes data-depending quatnization, whereas GPTQFrantar et al. (2022), AWQ Lin et al. (2023) or RPTQ Yuan et al. (2023) re-adjust the weights after quatnization to meet the dense model layers output. Quantization helps reduce memory transfer overhead for bandwidth-limited operations under small batch sizes. We pick GPTQ Frantar et al. (2022) as the representative approach. **Knowledge distillation** is a popular technique to bootstrap model performance using a stronger model Sanh et al. (2019); Wang et al. (2020c); Jiao et al. (2019); Sun et al. (2019); Passban et al. (2020); Li et al. (2021). Considering only the access to the original model, we opt for per-layer distillation as it has proven effective. **Efficient architecture design and NAS** provide another way to obtain a more efficient model. Manual design has been explored in Kitaev et al. (2020); Child et al. (2019); Wang et al. (2020b); Sun et al. (2020); Dai et al. (2020). In this work, we are constrained by the architecture of the teacher model. NAS offers another method to attain a model with an accuracy-latency trade-off Chen et al. (2020); Dong et al. (2021); Wang et al. (2020a); Xu et al. (2021); Yin et al. (2021). Unfortunately, the high cost of these NAS techniques limits their application to LLMs. We opt for fast NAS techniques applied to convolutional neural networks, such as DONNA Moons et al. (2021) and LANA Molchanov et al. (2022). We analyze, extend, and apply these techniques to LLMs, demonstrating the first scalable NAS method for LLMs.

## 3 METHOD

Our approach, depicted in Figure 2, comprises two main parts: 1) Candidate training (see Section 3), where layer-wise knowledge distillation is applied to train various candidate operations to mimic those in the original teacher model; 2) Architecture search phase, which explores the candidate space to find a model meeting custom constraints using linear optimization.

**Candidate training phase.** Our research focuses on Transformer-like models, particularly on enhancing the performance of layers within Transformer blocks. While embedding and output layers constitute a significant portion of the model, we defer their performance optimization for future work. Within each Transformer block, our focus is on optimizing two key layers: Attention and Multilayer Perceptron (MLP).

We define the set of student operations for each of the 2 layer types, offering a diverse range of candidates for each layer. The only constraint is that the input and output dimensions must match those of the corresponding teacher layer. Unlike other NAS approaches that aim to discover entirely new models, our goal is to replace the original model with a more efficient one. This confines our search to the architecture space "near" the teacher network. To achieve this, we conduct layer-wise knowledge distillation training to simulate the corresponding teacher layer.

Thus we train each student layer to simulate the corresponding teacher layer using layer-wise knowledge-distillation: $min \sum_{x \in X^D} L(t_i(x_i), s_{ij}(x_i))$ where $X_i^D$ is a set of training samples and $x_i$ is an input for the layer $i$, $L$ is a loss function.

We experimented with various layer-wise loss functions, including Mean Squared Error (MSE), cosine similarity, $L_1$ norm or various linear combinations of the losses. However, we have found that the best performing loss function in terms of its effect on accuracy of the distilled layer is the normalized MSE (SquareHead Kurtic et al. (2023b)): $L(a, b) = MSE(a, b)/MSE(a, 0)$. Further in the paper, we represent the results for this type of distillation loss.

In such candidate pre-training we can break down the training process into $(MLP\_ops + ATT\_ops) * layers$ independent minimization problems, allowing us to train all candidates simultaneously.

**Search phase.** In the rest of the paper, we adopt the following notation. We use $\mathbf{W} = w_{i,j}$ as a set of weights of all pre-trained students. We use a set of binary vectors $\mathbf{Z} = \{\mathbf{z}_i\}, i = 1..2N$, where $\mathbf{z}_i = 0, 1^{S^M}$ (or $z_i = 0, 1^{S^A}$ depending on layer type) is a one-hot vector representing the choice of the candidate layer. Thus, the candidate architecture defined by $\mathbf{Z}$ can be written as $\mathcal{C}(x; \mathbf{Z}, \mathbf{W})$. The usual formulation of the NAS problem can be expressed as:

$$\min_{\mathbf{Z}} \min_{\mathbf{W}} \sum_X \mathcal{L}(\mathcal{C}(x; \mathbf{Z}, \mathbf{W}), y) \tag{1}$$

The possible extensions of the problem may include various constraints like $\mathcal{F}(\sum_{i=1..2N} \mathbf{p}_i \mathbf{z}_i, \mathcal{P}) \geq 0$ where $p_i$ is a performance metric vector (e.g. latency, number of parameters) for each candidate operation of the $i$-th layer. The function $\mathcal{F}$ and user-provided constant $\mathcal{P}$ define the performance budget.

Similar to the LANA Molchanov et al. (2022) approach, we approximate loss function in the NAS problem as:

$$\sum_X \mathcal{L}(\mathcal{C}(x; \mathbf{Z}, \mathbf{W}), y) \approx \sum_X (T(x), y) + \sum_{i=1...2N} \mathbf{e}_i \mathbf{z}_i. \tag{2}$$

The first component represents the teacher training loss and remains constant. The second component is determined by $\mathbf{e}_i$ - vector of model loss changes if $i$-th layer. Here, $e_{i,j}$ is a difference between teacher loss and loss of the model where $i$-th layer is replaced with $j$-th (pre-trained) candidate on the same data and all the other layers in the model are left intact. Such a linear approximation lets us frame the NAS problem in terms of an ILP. This reformulation allows us to scale up to a large number of operations per layer and results in multiple diverse solutions.

The approximation shares similarities with the first-degree Taylor expansion of the student loss grounded on the premise that the teacher resides within the same search space. In other words, we approximate the non-linear effects of the layer-wise model changes with a linear function. Despite the fact that this approach neglects the cumulative effects of layer changes affects on model accuracy, empirical experiments validate its effectiveness.

Formally, the $k$-th solution, denoted as $\mathbf{z}^k$ is obtained by solving:

$$\min_{\mathbf{Z}} \sum_{i=1..2N} \mathbf{a}_i \mathbf{z}_i^k, \text{s.t.} \mathcal{F}(\sum_{i=1..2N} \mathbf{p}_i \mathbf{z}_i, \mathcal{P}) \geq 0, \tag{3}$$

where $\mathbf{z}_i$ is a one-hot vector. To improve the diversity of the solution, we add the following constraint:

$$\sum_{i=1..2N} \mathbf{z}_i^k \mathbf{z}_i'^{\hat{k}} \leq \mathcal{O}, \forall \hat{k} < k \tag{4}$$

The scalar $\mathcal{O}$ sets the maximum overlap of the suggested architectures, which we set as in (Molchanov et al., 2022) to be equal 0.7 of the total number of layers. In the ILP we minimize the linear approximation of model output changes due to layer replacement meeting the budget criterion and with overlap constraint we force the solver to provide varied solutions.

**Architecture Selection and Fine-tuning.** After the search phase, multiple architectures meeting the constraint are received and ranked based on the cumulative accuracy metric of per-layer operators. However, this ranking may not fully reflect the final model's accuracy, and models may react differently to further training. To address this, each architecture undergoes short training (100s millions of tokens), and their performance is evaluated (see Stage 3 in Figure 2). This approach helps identify the model that responds better to training, as detailed in Section 5.1. Once the best model is identified, a relatively short (tens of billions of tokens) version of the baseline pre-training is conducted, referred to as fine-tuning in our paper. The same learning rate schedule and dataset are used, with a higher learning rate for models with lower performance constraints. This choice is justified by the smaller and more efficient models having lower quality initially and needing larger optimization steps for improvement.

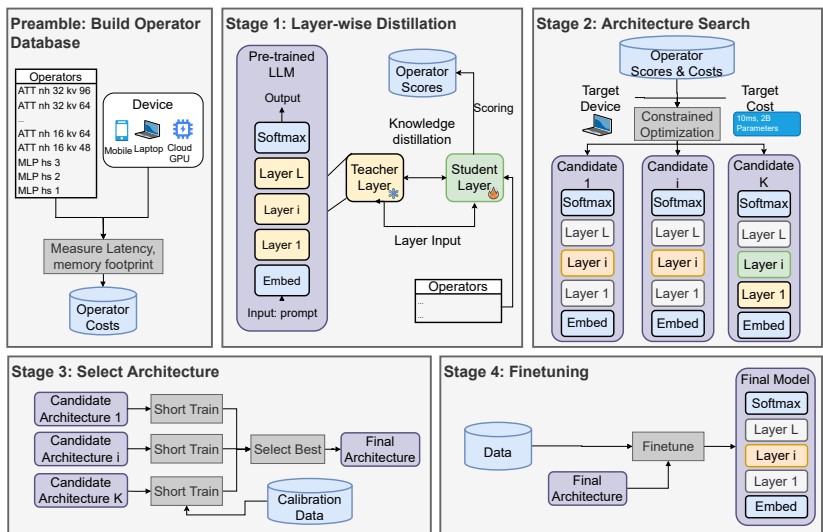

Figure 2: Method overview. In a preliminary stage, we select candidate operators and measure their cost (device-specific latency, memory footprint). (1) Each layer in the pre-trained model is distilled into suitable candidate operators by minimizing activation distance. Scores and weights for each teacher/operator pair are stored. (2) Under user-specified latency and memory constraints, we select the Top-K candidate architectures based on aggregate operator scores. (3) Each candidate architecture is fine-tuned on a calibration dataset, and the best architecture is selected. (4) Finally, the best architecture is fine-tuned to produce the UNAST model.

# 4 EXPERIMENTAL SETTINGS AND DETAILS

## 4.1 GENERAL SETUP

We focus on improving efficiency of Transformer-based models. Specifically, we gather insights by exploring 3 pretrained models: proprietary GPT-architecture models with 2B (GPT-2B) and 8B (GPT-8B) parameters, respectively, as well as the open LLaMa2-7B model. GPT-2B and GPT-8B are trained on 1.1T tokens datasets. GPT-2B and GPT-8B contain 24/32 layers with hidden dimension 2048/4096 and 16/32 heads in attention layers. The LLaMa2-7B models is the publicly available 7B GPT model trained on 2T tokens. It has similar configuration parameters as GPT3-8B. We used a version of LlaMa2-7B, shortly ($\sim$ 10B tokens) finetuned on a 1.1T tokens dataset. This was done due to the requirements of knowledge distillation, as this stage UNAST relies on the dataset used. It means that in the baseline results of the LlaMa2-7B, we use the locally-fine-tuned version of LlaMa2-7B.

**Implementation details.** We integrated the UNAST pipeline into the Megatron Nvidia (2024) codebase. For GPT experiments a 1.1T dataset was used for local distillation (5K steps, 2.5M tokens), evaluation (50 steps), finetuning (10K steps, 5M tokens). For LLaMa2 experiments a 3.5T dataset was used for distillation (5K steps, 5M tokens), evaluation (50 steps), finetuning (10K steps, 10M tokens). We use NVIDIA A100 as our target hardware. Training parameters are described in Appendix A.

**Evaluation metrics.** We use several model evaluation metrics in our experiments. *LM score* is the evaluation score after training on 50 samples on our 1.1T dataset. *LM-eval average* is computed over the zero-shot lm-harness benchmark Gao et al. (2023) accuracies. Hellaswag Zellers et al. (2019), Lambada Paperno et al. (2016), PiQA Bisk et al. (2019), RACE Lai et al. (2017) and Winogrande Sakaguchi et al. (2019) datasets are used, unless stated otherwise. Wikitext-103 Merity et al. (2016) is used to measure perplexity.

**Model size computation.** As one of the evaluation metrics, we will measure the size required to store the model, we assume BF16 format for the original model. Details can be found in the Appendix D.

## 4.2 UNAST DETAILS

**The cost - total training time.** We work with the GPT-3 8B and LLaMa2 7B models, originally trained on 1.1 trillion and 2 trillion tokens, respectively. In the local distillation step, we select 15 model candidates and fine-tune each for 10 billion tokens. The architecture search requires quick fine-tuning of 100 architectures, each for 102 million tokens. Finally, the best architecture is fine-tuned for 20 billion tokens. Each of the first two stages can be parallelized. In a sequential implementation, creating a single model incurs only 9% of the original costs (an 11× speedup), or 1.5% with parallelization. To produce three sub-models, costs could be reduced by 25× or even 85× with parallelization.

> *Finding 1.* Our approach reduces training time by $11\times$ - $85\times$ to get 3 additional models.

**Candidates.** We establish a set of candidate operations based on the model and layer types, focusing on Attention or MLP layers with inputs and outputs of size $N$. The following operations can be applied: ***Teacher operation***: A copy of the corresponding baseline layer, requiring no knowledge distillation. ***Identity layer***: Essentially a skip-layer, propagating the input of the block to the output, also not requiring knowledge distillation. ***Linear layer***: Replacing an Attention or MLP block with a linear layer of size $N \times N$. This candidate, along with the Identity layer, is the most computationally efficient but may significantly affect model accuracy. Some of the candidates are layer-type dependent, meaning they can only replace either Attention or MLP layers, representing the same layer type with different parameters. The first such candidate is ***downsized Attention layer*** with lower number of heads or number of channels. Here, we do layer-wise distillation from the teacher Attention layers to the candidates with lower parameters. The symmetrical candidate for MLP is a ***downsized MLP layer*** with lower hidden expansion.

**Knowledge distillation.** The success of the UNAST approach depends on the quality of candidates. To get strong candidates we perform per-layer distillation with the goal of mimicking teacher layers. We pre-train separate candidates for Attention and ML layers. In Figure 3 we show the dependency of LM score on latency for various candidates. In this evaluation we replace all teachers of the corresponding type with the pre-trained layer candidates.

UNAST necessitates estimating per-candidate scores regarding how well they mimic their teacher. Detailed ablations led us to consider the Mean Squared Error (MSE) between teacher and candidate activations as a score. We showcase per-layer scores for various candidates in Figure 4.

**Knowledge distillation loss.** We ablate multiple loss functions for the per-layer distillation in the Table 1. The metric is global MSE for the layers in the head, middle, tail of the model body. Total model row represents LM score of the model where all layers of the corresponding type are replaced with the distilled candidates. Attention layer candidate: attention layer with 16 heads and 128 channels. MLP candidate: MLP layer with expansion factor 2. We conclude that SquareHead Kurtic et al. (2023b) allows the students to better track the teacher, as shown by the lower MSE loss, and leads to better models overall, as shown by the better LM score.

We can see that MLP layers are **more sensitive** to down-sizing than Attention layers, as an equivalent removal of channels in the Attention layer damages the model accuracy slightly less than MLP. As expected, the most sensitive layer is the last one as it has the highest impact on the model output and there are no layers after it to recover the possible inaccuracies.

**Performance metric.** A key performance metric we aim to improve in UNAST is latency. This metric naturally depends on various deployment settings, such as the regime in which a model is used or its parameters (e.g. batch size or input sequence size). Our approach works with any deployment setting. We build a timing lookup table for any setup, which is then used in the search phase. We note that it does not require any re-training of the candidates. We argue that the main body (Transformer blocks) consumes most of compute in the pre-fill stage. We analyse costs in Appendix B.

**Lookup Table.** Accurate latency estimation is crucial for model performance optimization, as the resulting model depends on the measurement regime. It's essential to assess latency in the deployment regime. Our experiments measure latency only in the pre-fill regime. We leave the auto-regressive regime for future work. A hardware-specific lookup table is computed in the pre-fill phase with batch size 1 and sequence length 4096.

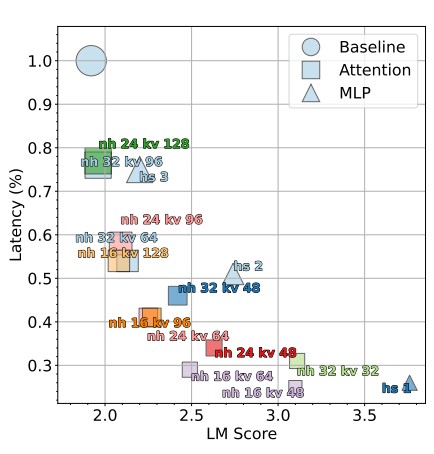

Figure 3: Latency vs LM loss trade-off of different candidates used in UNAST. Size represents the relative ratio of the parameter count.

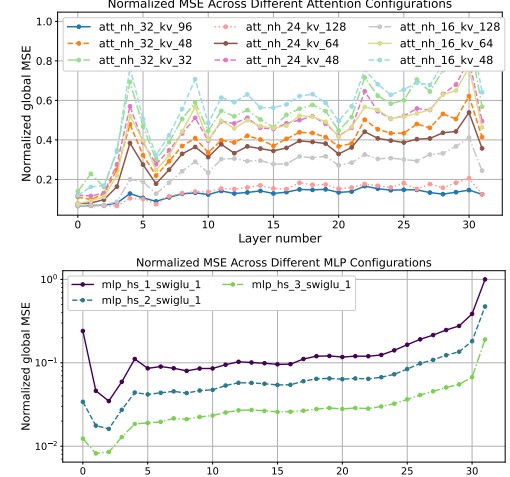

Figure 4: Per-layer normalized MSE loss induced by replacing a single teacher layer with specified operator.

| Method | L1 Loss↓ | L2 Loss↓ | SquareHead↓ |
|---|---|---|---|
| Attention, head | 2.9e-3 | 2.8e-3 | **2.5e-3** |
| Attention, middle | 1.14e-2 | 1.15e-2 | **1.02e-2** |
| Attention, tail | 1.20e-2 | 9.0e-3 | **8.9e-3** |
| Total model ATT, LM score | 2.120 | 2.098 | **2.094** |
| MLP, head | 2.12e-2 | 1.70e-2 | **1.65e-2** |
| MLP, middle | 2.67e-2 | **2.63e-2** | 2.64e-2 |
| MLP, tail | 2.38e-1 | 2.30e-1 | **2.29e-1** |
| Total model MLP, LM score | 2.790 | 2.760 | **2.750** |

Table 1: Study of different knowledge-distillation losses. We report the teacher-student MSE for each loss, as well as the resulting LM score.

| Pool | Budget | LM-eval↓ | Wiki ppl ↓ |
|---|---|---|---|
| Baseline | 100% | 63.46 | 7.28 |
| Only Attention | 80% | 62.01 | 7.49 |
| Only MLP | 80% | 61.84 | 7.78 |
| Only Linear | 80% | 60.78 | 7.89 |
| Only skip | 80% | 60.34 | 8.18 |
| Full pool | 80% | 62.31 | 7.44 |

Table 2: Ablation results with limited pool of operators. Evaluation results for the UNAST architectures built from various partitions of the candidates pool.

# 5 EXPERIMENTAL RESULTS

| Compression | Candidate Architecture Finetuning | Final Validation Perplexity↓ |
|---|---|---|
| 33% | No | 2.57 |
| 33% | Yes | 2.41 |
| 50% | No | 2.32 |
| 50% | Yes | 2.25 |

Table 3: Evaluation of the impact of running short fine-tuning on architecture candidates before selecting the final architecture.

## 5.1 ABLATION STUDIES

We study various candidate operations (layers) in more details in this section. Particularly, we are interested in ablating compression with: (i) only Attention candidates; (ii) only MLP layers; (iii) identity candidates (layer skip); (iv) linear layer candidates (cheap linear layer instead of original layers); (v) all candidate layers with the full pool. We target budgets 33%-80%. All ablations are summarized in Table 2. The full pool of candidates shows the best overall result.

| Model | Budget | ARC-E | LAMBADA | PIQA | WinoGrande | Average | wiki-103 | Non-emb params, B | Through- put |
|---|---|---|---|---|---|---|---|---|---|
| GPT3 8B | 100% | 73.4 | 70.4 | 78.0 | 69.6 | 72.9 | 7.28 | 6.4 | 3060 |
| GPT3 8B | 80% | 71.9 | 69.9 | 77.0 | 67.6 | 71.6 | 7.44 | 5.5 | 3764 |
| GPT3 8B | 75% | 69.9 | 68.7 | 77.2 | 68.0 | 71.0 | 7.66 | 5.0 | 3960 |
| GPT3 8B | 60% | 69.4 | 67.0 | 77.1 | 65.7 | 69.8 | 7.99 | 4.4 | 4455 |
| GPT3 8B | 50% | 68.7 | 65.5 | 76.9 | 67.5 | 69.6 | 8.10 | 3.7 | 4961 |
| GPT3 8B | 33% | 63.9 | 58.1 | 76.0 | 59.3 | 64.3 | 9.26 | 2.4 | 5992 |
| GPT3 8B depth | 50% | 66.7 | 63.2 | 75.3 | 63.1 | 67.1 | 10.3 | 3.2 | 5520 |
| GPT3 2B | 100% | 63.2 | 60.9 | 75.9 | 61.7 | 65.4 | 9.35 | 1.2 | 7421 |
| GPT3 2B | 50% | 56.6 | 48.7 | 72.8 | 52.6 | 57.7 | 11.5 | 0.7 | 10915 |
| GPT3 2B | 33% | 54.8 | 38.8 | 70.5 | 51.4 | 53.8 | 13.4 | 0.5 | 13479 |
| GPT3 843M | 100% | 53.2 | 50.2 | 70.0 | 54.3 | 56.9 | 12.4 | 0.3 | 11692 |

| Model | Budget | ARC-E | LAMBADA | PIQA | WinoGrande | MMLU 5 shot | HellaSwag 10 shot | Non-emb params, B | Through- put |
|---|---|---|---|---|---|---|---|---|---|
| LLaMa 7b Touvron et al. (2023) | 100% | 71.0 | 67.6 | 78.1 | 68.4 | 45 | 76.2 | 6.7 | 3138 |
| OpenLLaMa-7Bv2 Geng & Liu (2023) | - | 69.5 | 63.8 | 79.9 | 66.0 | 40 | 76.6 | 7.0 | - |
| Pythia-6.9B Biderman et al. (2023) | - | 60.2 | 47.1 | 75.2 | 59.9 | 26 | 64.4 | 6.4 | - |
| LLaMa 7b - UNAST | 75% | 72.7 | 68.2 | 76.8 | 67.8 | 38 | 75.1 | 5.3 | 3775 |
| Compresso Guo et al. (2023) | - | 66.0 | - | 72.9 | 63.4 | 26 | - | 4.5 | - |
| LLM-Pruner Ma et al. (2023a) | - | 59.2 | - | 73.4 | 63.4 | 24 | 56.5 | 4.5 | - |
| LLaMa 7b - UNAST | 60% | 68.5 | 65.8 | 74.5 | 66.5 | 45 | - | 4.2 | 4423 |
| Pythia-2.8B | - | 57.9 | 50.1 | 73.8 | 58.6 | 27 | - | 2.5 | - |
| OpenLLaMa-3Bv2 | - | 63.7 | 59.1 | 78.1 | 63.3 | 26 | - | 3.2 | - |
| ShearedLLaMa-2.7B | - | 67.0 | 68.4 | 75.8 | 64.2 | 26 | - | 2.5 | - |
| LLaMa 7b - UNAST | 40% | 63.5 | 55.8 | 72.4 | 61.0 | 38 | - | 2.8 | - |

Table 4: Evaluation results for various baseline and UNAST models. Non-embedding parameter count stands for number of parameters in the backbone, without embedding and output layers. LM-harness average: ARC-easy, LAMBADA, PIQA and WinoGrande in zero-shot. For techniques other than ours, we report numbers from the original or overview papers, we do not replicate them.

> ***Finding 2.*** Layer skipping under full model finetuning exhibits the poorest performance; replacing an entire layer with a simple linear one yields better results.

> ***Finding 3.*** Opting for smaller alternatives, such as reducing the number of heads or the MLP expansion factor, proves to be the most effective strategy.

**Architecture evaluation step.** Recall that, during ILP, we identify 100 architectures and rank them by a score proxy, which does not fully represent the model's performance. We propose fine-tuning these models for a minimal 100M tokens, corresponding to 50 training iterations, and re-ranking them based on the final LM loss. To evaluate this step's importance, Figure 5 shows the correlation between LM loss before and after fine-tuning. We observe no correlation in ranking, underscoring the critical nature of this step for UNAST. Similarly, Figure 3 shows the importance of running fine-tuning before selecting the final architecture. The original LANA algorithm omits this step.

## 5.2 DISCUSSION OF COMPRESSION RESULTS

**Depth pruning.** Depth pruning is popular for its simplicity, relying on importance estimation of blocks, followed by trimming and finetuning. We implement the Shortened LLaMA method Kim et al. (2024) at the layer level, comparing it with UNAST. For UNAST, we use two set- tings: (i) the same scoring technique with ILP search and architecture evaluation, and (ii) the full UNAST pipeline. In (i), there's no local distillation, making it a simple improvement on depth pruning.

Implementing this is straightforward, as we can create a pool with two candidates: the teacher layer and the identity layer. The ILP provides multiple architecture candidates, with the best

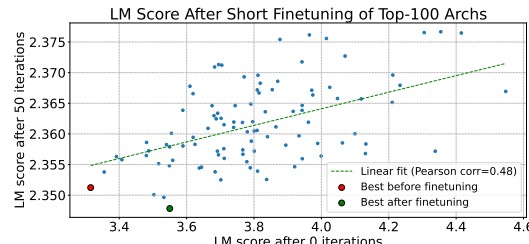

Figure 5: Architectures evaluation results for GPT 8B, 50% latency budget. X axis represents lm score of an architecture without fine-tuning, Y axis - lm score of the architecture after 50 steps (25k samples) of fine-tuning.

in terms of cumulative accuracy (Mean Squared Error) chosen for finetuning. We see that the ILP improves LM eval from 47.9 to 57.8 (see comparison against ShortenedLlama in Appendix C). Full UNAST produces better final architecture with LM eval at 60.3.

| Method | GPT3-8B | | | | | Method | LLaMa7B | | | | |
|---|---|---|---|---|---|---|---|---|---|---|---|
| | Size GB | Wiki103 PPL | LM eval | Speed up | Param | | Size GB | Wiki103 PPL | LM eval | Speed up | Param |
| Baseline | 15.9 | 7.28 | 63.4 | 1.0x | 6.4 | | 12.55 | 5.48 | 62 | 1.0x | 6.7 |
| Quantization* | | | | | | Quantization* | | | | | |
| GPTQ Frantar et al. (2022) 3b | 6.10 | 11.8 | 32.3 | - | 6.4 | GPTQ 3b | 2.75 | 8.13 | 47.3 | - | 6.7 |
| UNAST 50% + 4b | **5.63** | **9.44** | **57.8** | **1.6x** | **3.7** | U60% 4b | **2.33** | **6.66** | **58.8** | **1.4x** | **4.5** |
| GPTQ 4b | 6.80 | 9.31 | 51.6 | - | 6.4 | GPTQ 4b | 3.50 | 5.78 | 60.4 | - | 6.7 |
| RTN Dettmers et al. 4b | 6.80 | 16507 | 30.1 | - | 6.4 | RTN 4b | 3.50 | 10557 | 32.1 | - | 6.7 |
| UNAST 75% + 4b | **6.23** | **9.06** | **58.1** | **1.3x** | **5.0** | U75% 4b | **2.94** | **6.01** | **60.8** | 1.2 | 5.0 |
| Sparsity | | | | | | Sparsity | | | | | |
| SparseGPT Frantar & Alistarh (2023) 1:4 | 6.90 | 1427 | 30.2 | 1.4x | 1.6 | SparseGPT 1:4 | 3.50 | 1262 | 30.3 | 1.4x | 1.6 |
| SparseGPT 25% | 6.90 | 246 | 31.5 | - | 1.6 | SparseGPT 25% | 3.50 | 224 | 31.4 | - | 1.6 |
| UNAST 50% s50% | 7.36 | **9.36** | **58.0** | **1.6x** | 1.9 | U60% s40% | 3.64 | 7.83 | 55.0 | 1.4x | 1.6 |
| SparseGPT 2:4 | 9.90 | 12.0 | 54.7 | 1.4 | 3.2 | SparseGPT 2:4 | 6.50 | 224 | 52.6 | 1.4x | 3.2 |
| SparseGPT 50% | 9.90 | 8.87 | 60.3 | - | 3.2 | SparseGPT 50% | 6.50 | 6.90 | 58.7 | - | 3.5 |
| Depth pruning Kim et al. (2024) | 10.38 | 11.70 | 47.9 | 1.7 | 3.5 | - | - | - | - | - | - |
| UNAST 50%_d | 9.88 | 10.33 | 57.8 | 1.7 | 3.2 | U60% | 8.34 | 6.38 | 59.4 | 1.4x | 4.2 |
| UNAST 50% | 10.80 | **8.10** | **60.3** | **1.6x** | 3.7 | U75% s50% | 5.39 | 6.01 | **61.1** | 1.2x | 2.6 |

Table 5: Comparison of UNAST with SOTA post-training compression techniques. UNAST allows to apply the same compression methods on top of the resulting architectures resulting in even more succinct models. GPT 8b. * We assume no speed up from quantization to 3 or 4 bits, for lack of suitable hardware implementations. SliceGPT model includes finetuning. Speed up is computed as a ratio to the base model in BF16 with BS=32, input/output length is 16/512 tokens. The speed up from UNAST+Ampere 2:4 is not supported at this time and we only report the speed up from UNAST. UNAST 50%_d stands for UNAST with layer skipping as candidates only, sX% for unstructured sparsity with X%.

**Memory footprint target.** UNAST also supports minimizing the final number of parameters. In Appendix F we compare the performance of these models. We find that optimizing for fewer parameters improves latency but slightly reduces accuracy compared to latency-targeted architectures.

**Full model compression.** Main results of compressing GPT3-2B, GPT3-8B and LLaMa2-7B as well as comparisons to other models and efficiency methods are summarized in Table 4. From these comparisons, it is clear that UNAST enables the creation of multiple models constrained to different target latencies, while maintaining reasonable accuracy reductions. The results show that UNAST produces efficient models that outperform other models, such as OpenLLaMa and Pythia, in terms of accuracy, despite having fewer parameters.

## 5.3 COMBINING WITH OTHER PTO TECHNIQUES

UNAST is a new Post-Training Optimization (PTO) approach, orthogonal to popular techniques like quantization and sparsity. In this section, we illustrate that these techniques complement each other well. Furthermore, combining UNAST with quantization/pruning yields markedly superior results compared to using a single technique. Specifically, UNAST focuses on replacing layers with more efficient candidates emulating the behavior of the original layer, while quantization reduces the bit-width of model parameters and sparsity removes redundant parameters. As shown in Table 5, combining UNAST with quantization or pruning results in significantly improved accuracy and efficiency compared to models of the same size compressed using a single technique. For instance, the UNAST 50% + 4 bits quantization for the GPT 8B model yields a smaller resulting model size than the baseline compressed to 3 bits, with higher accuracy.

> *Finding 4.* Combining UNAST with other PTO techniques is significantly better than individual PTO techniques.

## 6 DISCUSSION ON RESULTING MODEL STRUCTURES

In this section we describe the insights on the model structure we can infer from the UNAST resulting models. We focus on GPT 8B model.

**General Observations.** First, we discuss the models with restricted latency budget and the layer operator candidates are taken from all available distilled candidates. The figures with structures of the resulting models are presented in Appendix G. From the model structures we can make the following observations: 1. The method tends to compress Attention layers rather than MLP, and 2.

The algorithm prefers to achieve speedup by removing channels rather than Attention heads. However, reducing the number of heads or channels has a similar effect on latency: the latency of the layer with 16 heads and 128 channels is equal to the latency of the layer with 32 heads and 64 channels. This means that reducing the number of heads affects accuracy more than reducing channels.

This finding is confirmed for other budget latency models as well. (See Appendix G).

> *Finding 5.* Compressing Attention layers has a lower effect on model accuracy than MLP compression.

> *Finding 6.* Opting for a reduction in the number of heads rather than number of channels in the Attention layer has a positive effect on the model accuracy.

**Attention vs MLP.** To delve deeper into the impact of compression on the two types of basic layers within Transformer models, we conducted experiments where we restricted the search of potential candidates to one type of layer: Attention or MLP, while keeping all other layers the same as the teacher. This approach forces UNAST to achieve speedup solely from one type of layer. The accuracy results are presented in Table 2. For model structure illustrations, please refer to Appendix G, reinforcing our assertion about the importance of heads in the attention layer over channels. From the results of isolated MLP layer compression search, we observe that layers in the first half of the model body are more readily compressed than those in the second half. A similar trend is noticed for Attention layers. However, UNAST decides to apply high compression on the last 5 MLP layers.

> *Finding 7.* Layers in the first half of the model are more amenable to compression compared to those in the second half.

**Depth pruning.** Next we study the affect of skipping the layers, where the evaluation results are presented in the Table 4. We can see (Appendix G) that it is preferable to remove more attention layers rather than MLPs. Moreover, it chooses to remove the layers in the middle of the model and keep a large group of MLP layers in the tail.

> *Finding 8.* Depth pruning tends to prune both types of layers around the middle of the model.

**Relationship to LANA** LANA Molchanov et al. (2022) explores layer-wise distillation by focusing on CNN architectures in vision. Our work extends this study to the transformer architecture and language models. This involves studying different sets of operators, including attention and MLP. Besides, the scale of LLMs imposes implementation constraints and the use of tensor and pipeline parallelism. Our work studies different pre-training losses, which were not discussed in the LANA work. An additional contribution of our work is the introduction of a short fine-tuning stage for candidate architectures. The experimental results in Table 3 highlight the advantages of this approach.

## 7 CONCLUSION AND LIMITATIONS

In this paper, we introduce a novel NAS approach for Transformer-based Large Language Models, facilitating adaptable model tuning to accommodate various performance and resource constraints. UNAST enhances both performance and efficiency, making it essential for creating high-performing, resource-efficient models. Our analysis of layer sensitivity to compression—especially structured pruning—identifies components prone to degradation, guiding the development of effective compression strategies.

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

Table 6: Hyperparameters for Knowledge-Distillation

| Parameter | Value |
|---|---|
| Optimizer | Adam |
| Global batch size | 256 |
| Momentum | 0.0 |
| Max/Min LR | 5e-4/5e-5 |
| LR decay | Cosine |
| Iterations | 5k |

Table 7: Hyperparameters for Fine-tuning

| Parameter | Value |
|---|---|
| Optimizer | Adam |
| Global batch size | 256 |
| Momentum | 0.1 |
| Max/Min LR | 5e-4/5e-5 |
| LR decay | Cosine |
| Iterations | 10k |

Zhewei Yao, Reza Yazdani Aminabadi, Minjia Zhang, Xiaoxia Wu, Conglong Li, and Yuxiong He. Zeroquant: Efficient and affordable post-training quantization for large-scale transformers. In *Advances in Neural Information Processing Systems (NeurIPS)*, 2022.

Yichun Yin, Cheng Chen, Lifeng Shang, Xin Jiang, Xiao Chen, and Qun Liu. Autotinybert: Automatic hyper-parameter optimization for efficient pre-trained language models. In *Proceedings of the 2021 International Joint Conference on Natural Language Processing*, pp. 5146–5157. Association for Computational Linguistics, 2021. URL https://aclanthology.org/2021.ijcnlp-main.428.

Shuzhou Yuan, Ercong Nie, Bolei Ma, and Michael Färber. Why lift so heavy? slimming large language models by cutting off the layers. *arXiv preprint arXiv:2402.11700*, 2024. URL https://arxiv.org/abs/2402.11700.

Zhihang Yuan, Lin Niu, Jiawei Liu, Wenyu Liu, Xinggang Wang, Yuzhang Shang, Guangyu Sun, Qiang Wu, Jiaxiang Wu, and Bingzhe Wu. Rptq: Reorder-based post-training quantization for large language models. *arXiv preprint arXiv:2305.01693*, 2023.

Rowan Zellers, Ari Holtzman, Yonatan Bisk, Ali Farhadi, and Yejin Choi. Hellaswag: Can a machine really finish your sentence? *arXiv preprint arXiv:1905.07830*, 2019.

Mingyang Zhang, Hao Chen, Chunhua Shen, Zhen Yang, Linlin Ou, Xinyi Yu, and Bohan Zhuang. Loraprune: Pruning meets low-rank parameter-efficient fine-tuning. *arXiv preprint arXiv:2305.18403*, 2023.

## A    TRAINING PARAMETERS

In tables 6 and 7 we present the hyperparameters for knowledge distillation training and fine-tuning (final training of the best architecture) respectively. In the short training of the top-k candidates for their ranking we used the same hyperparameters as in the fine-tuning.

## B    MODEL PERFORMANCE METRICS

In table 8 we profile the inference of GPT3-8B model under two workloads: pre-fill and auto-regressive. We can see that most of the time in both regimes is taken by the Transformer blocks, more than 80%. However, in terms of parameter count main body takes 76%.

## C    DEPTH PRUNING

In Table 9 we compare our implementation of ShortenedLlamaKim et al. (2024) applied to GPT3 8B vs full UNAST pipeline. In our ShortenedLlama implementation we basically pick the best architecture in terms of sum of global MSEs and then run the finetuning. In UNAST on contrary we first run short fine-tuning and pick the best model based on evaluation and then start the finetuning for the chosen architechture. We can see than UNAST approach outperform the ShortenedLlama approach in accuracy and speedup.

| | Params, M | Latency, s | |
| --- | --- | --- | --- |
| | | prefill | zero-shot |
| Embedding | 1048 | 15% | 0% |
| Main body | 6425 | 85% | 99% |
| Output | 1048 | 0% | 1% |

Table 8: Performance metrics different stages of GPT3-8B model. The main body, transformer decoder blocks, consume 76% parameters and 94% compute and are selected for compression with UNAST.

| Method | Wiki-103 PPL | LM eval | Speedup |
| --- | --- | --- | --- |
| Depth pruning | 11.70 | 47.9 | 1.7 |
| UNAST depth | 10.33 | 57.8 | 1.7 |
| UNAST | 8.1 | 60.3 | 1.6 |

Table 9: Compressing the model to 50% budget with original Depth pruning, UNAST approach to depth pruning and full UNAST algorithm.

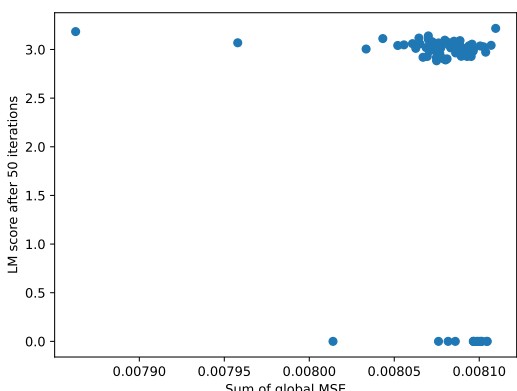

Figure 6: Evaluation of LLaMa 7B 60% latency budget model. X axis stands for the metric we use in ILP, lower is better. Y axis is lm score of a model after 50 steps of the finetuning.

## D    MODEL SIZE COMPUTATION

We compute the models sizes in table 5 the following way. The embedding and output layers are always uncompressed and stored in float16 precision. The main body parameters can be compressed in different ways. For the UNAST compression we compute the total number of main body parameters in the resulting architecture. For GPTQ compression we assume that each element takes certain number of bits. For SparseGPT we compute the number of non-zero parameters. For example, to compute model size of UNAST 50% + GPTQ 4b for GPT 8B model, we consider 2.1B embedding + output parameters, add 3.7B parameters compressed to 4 bits and as result we get (2.1B * 16 bits + 3.7B * 4 bits) = 5.63 GB. For UNAST 50% + SparseGPT 50%, the formula is following (2.1B + 3.7B * 0.5) * 16 bits = 7.36GB.

## E    EFFECT OF SHORT FINETUNING

To validate the effect of the short finetuning on the choice of the best model we compare the metric we use in the ILP - sum of the layer-wise global MSE against the evaluation score after 50 steps of the finetuning for each of 100 models that ILP outputs. The result is shown in Figure 6. We can see that the top-1 model after ILP might be in the cluster of the worst models after the finetuning. It means that short finetuning helps to properly rank the sampled models.

## F    PARAMETERS CONSTRAINT

UNAST supports various types of model efficiency targets. In Table 10 we compare latency and memory footprint targets. We can see that UNAST with memory footprint constraint results in more faster and succinct models which in its turn negatively affects the model accuracy.

| Objective | LM-eval | Wiki PPL | latency ratio | Params ratio |
|---|---|---|---|---|
| Baseline | 63.4 | 7.28 | 1.0 | 1.0 |
| 80% latency | 62.3 | 7.44 | .86 | .89 |
| 80% params | 62.3 | 7.52 | .83 | .85 |
| 50% latency | 60.3 | 8.10 | .62 | .68 |
| 50% params | 57.8 | 8.61 | .59 | .62 |

Table 10: Optimization objective ablation. Latency and number of parameters include uncompressed embedding and output layers.

## G  MODEL STRUCTURE

In Figures 7, 8, 9, 10 structure for Attention and MLP layers for GPT3 8B with constrained latency budget. In Figure11 we see the structures of the models which gain latency profits from only one type of the layers whereas the layers of the other type are same as in the teacher. We can see which layers UNAST suggests to remove in Figures 12 and 13. Figure 14 shows the structure of the model with constrained parameters count.

In Figures 15, 16 we can see structures of GPT3 2B model. Figures 17 and 18 represent UNAST resulting architectures of Llama 7B model with constrained latency budget.

Based on the figures, we can confirm the statements we did in the Section 6:

1. Attention layers are less sensitive to compression than MLP.
2. Reducing number of heads in Attention layer has less affect on the model accuracy than reducing number of channels
3. Layers in the first half of the model are easier to compress than the layers in the second half

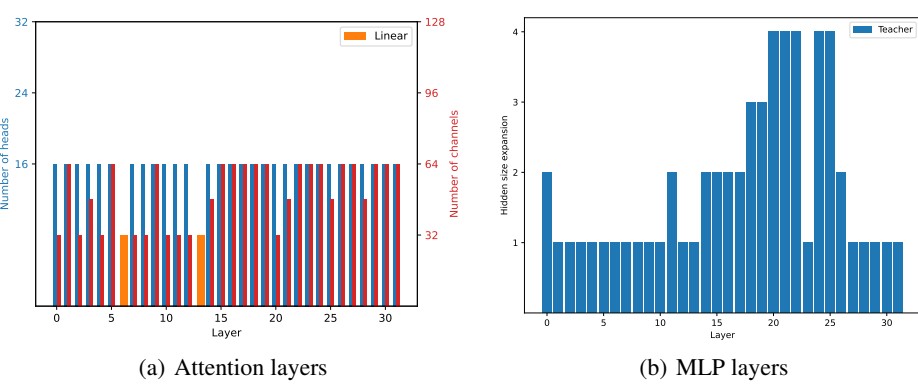

(a) Attention layers          (b) MLP layers

Figure 7: Layers parameters for a single architecture with latency budget 33%. Hatched bars stand for the layers that are identical to the corresponding teacher (baseline) layers.

## H  PRUNED LAYERS IN THE CANDIDATES POOL

As ablation study we created two possible candidates: 2:4 pruned layer and quantized layer. Using these candidates we can get some insights which layers are more sensitive to different types of compression. For that we pruned the original model using SparseGPT (2:4 sparsity) or GPTQ (with 4 and 8 bits). Then. for each Attention or MLP layer we created three candidates: 1. Candidate with compressed linear module (QKV or FC1) 2. Candidate with compressed second linear module (attention output or FC2). 3. Both linear modules are compressed. Then we evaluated each candidates the same way we do after the pretraining, in this case we didn't do any knowledge distillation.

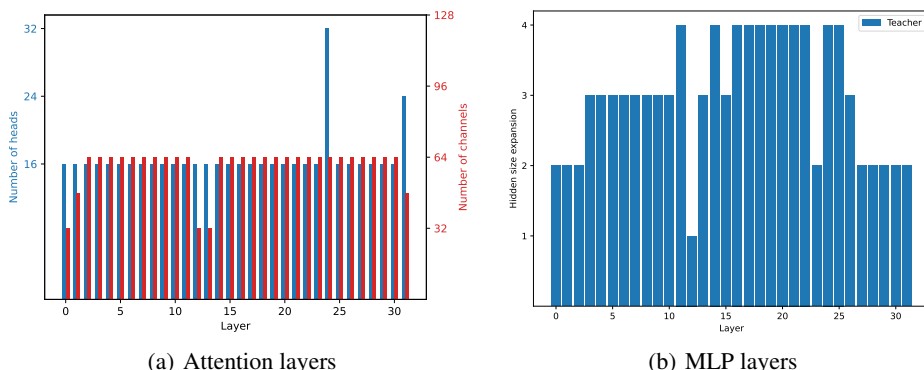

(a) Attention layers            (b) MLP layers

Figure 8: Layers parameters for a single architecture with latency budget 50%. Hatched bars stand for the layers that are identical to the corresponding teacher (baseline) layers.

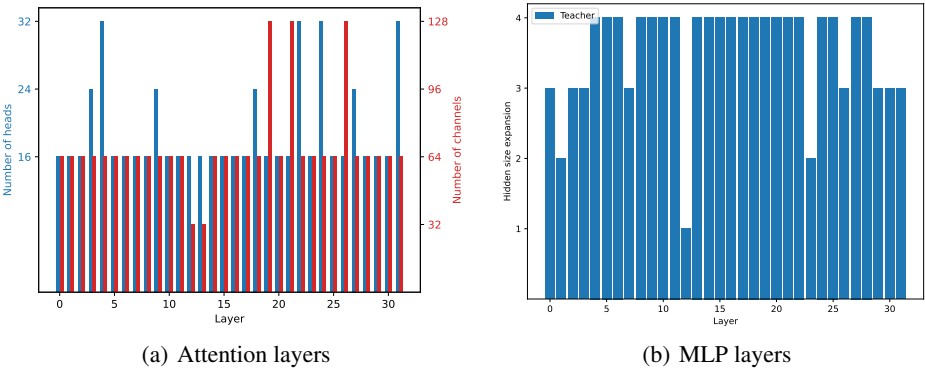

(a) Attention layers            (b) MLP layers

Figure 9: Layers parameters for a single architecture with latency budget 60%. Hatched bars stand for the layers that are identical to the corresponding teacher (baseline) layers.

We added each of the candidate operators to the original pool of candidates. Turns out that the best model (see Figure 19) was chosen to use mostly quantized candidates, and quantize both modules inside each of the layers. When we removed the quantize candidates from the pool to see the effect of pruned operations, the best model was designed like in Figure 20. Here, we can see that UNAST mostly decides to prune both modules in MLP layers but it prefers to use downsized candidates in case of Attention layers.

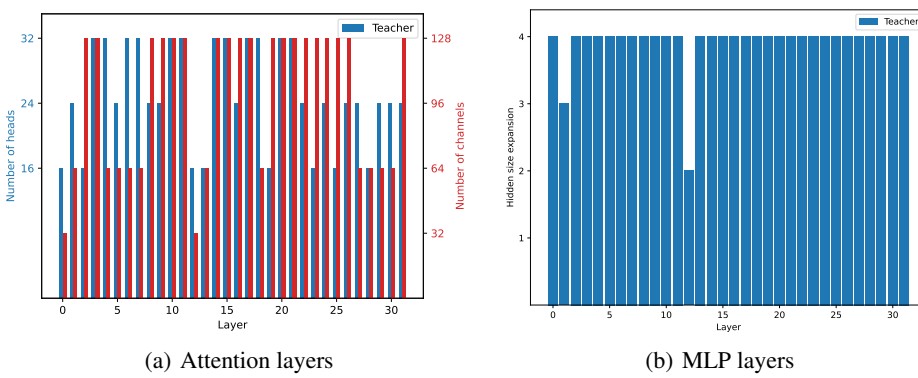

(a) Attention layers          (b) MLP layers

Figure 10: Layers parameters for a single architecture with latency budget 80%. Hatched bars stand for the layers that are identical to the corresponding teacher (baseline) layers.

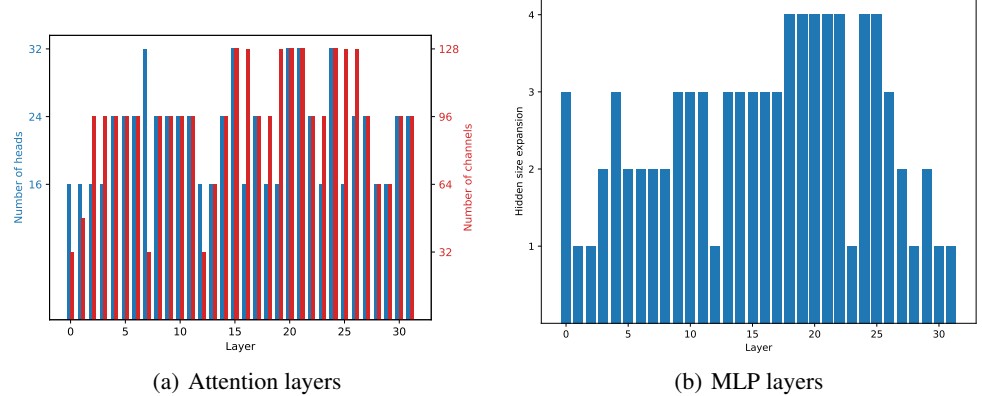

(a) Attention layers          (b) MLP layers

Figure 11: Layers parameters for 2 different result architectures with only Attention (left) and only MLP (right). The latency budget for both candidates is 80%. The layers of the other type are uncompressed. Hatched bars stand for the layers that are identical to the corresponding baseline layers.

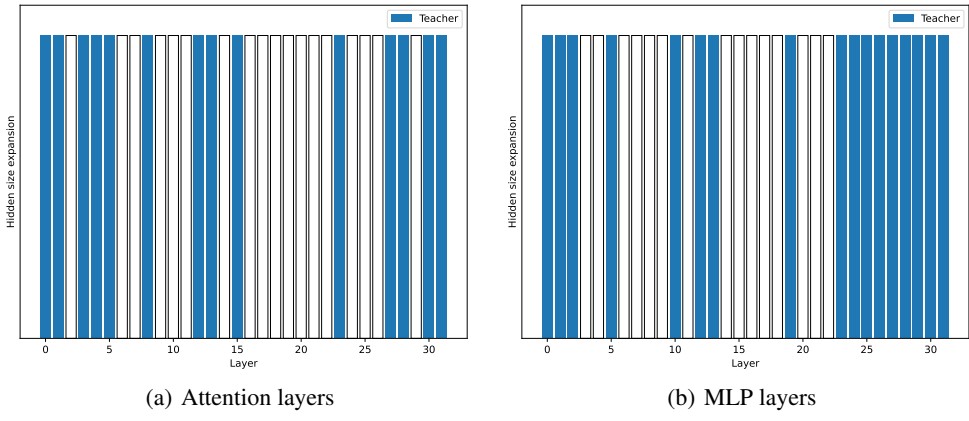

(a) Attention layers          (b) MLP layers

Figure 12: Layers of the UNAST architecture with 50% latency budget from skip layers pool.

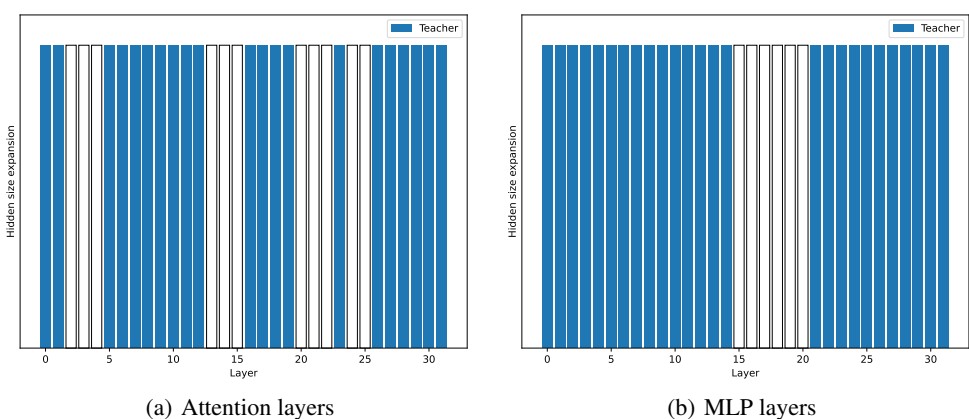

(a) Attention layers

(b) MLP layers

Figure 13: Layers of the UNAST architecture with 75% latency budget from skip layers pool.

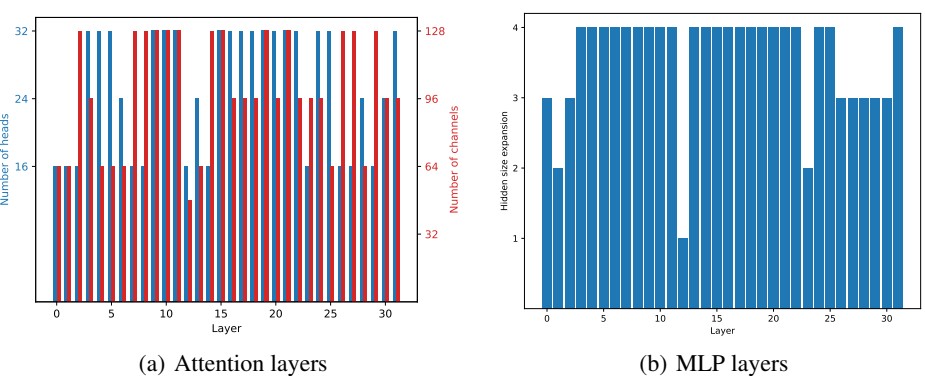

(a) Attention layers

(b) MLP layers

Figure 14: Layers parameters for a single architecture with parameters count budget 80%. Hatched bars stand for the layers that are identical to the corresponding teacher(baseline) layers.

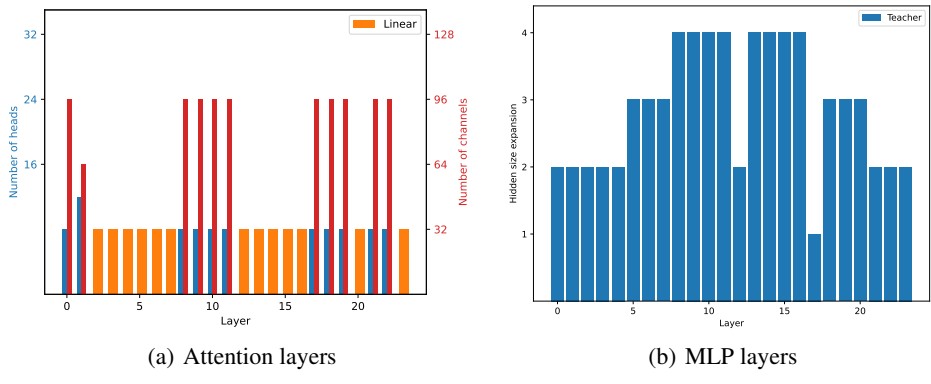

(a) Attention layers

(b) MLP layers

Figure 15: GPT-2B. Layers parameters for a single architecture with latency budget 50%. Hatched bars stand for the layers that are identical to the corresponding teacher(baseline) layers.

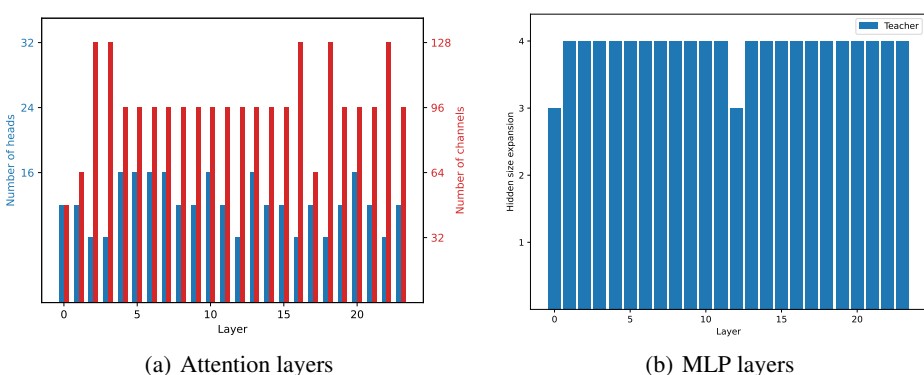

(a) Attention layers

(b) MLP layers

Figure 16: GPT-2B. Layers parameters for a single architecture with latency budget 80%.Hatched bars stand for the layers that are identical to the corresponding teacher(baseline) layers.

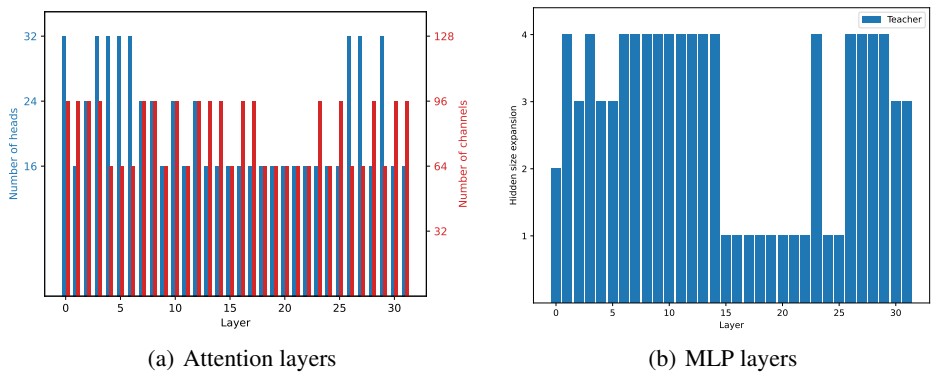

(a) Attention layers

(b) MLP layers

Figure 17: LLaMa 7B. Layers parameters for a single architecture with latency budget 60%.

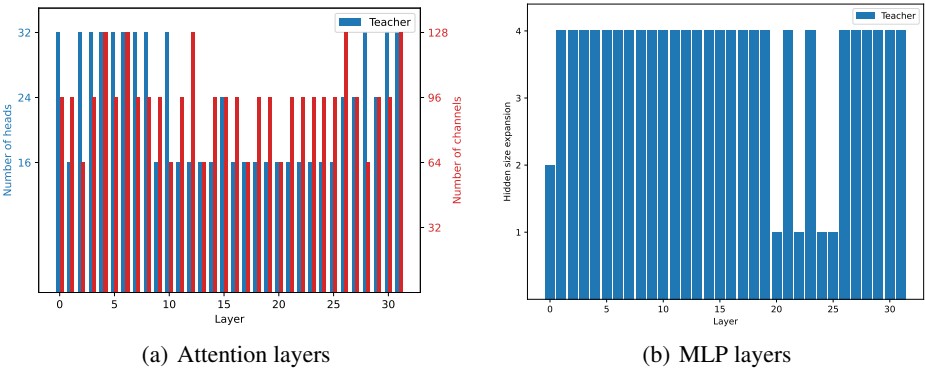

(a) Attention layers

(b) MLP layers

Figure 18: LLaMa 7B. Layers parameters for a single architecture with latency budget 75%.

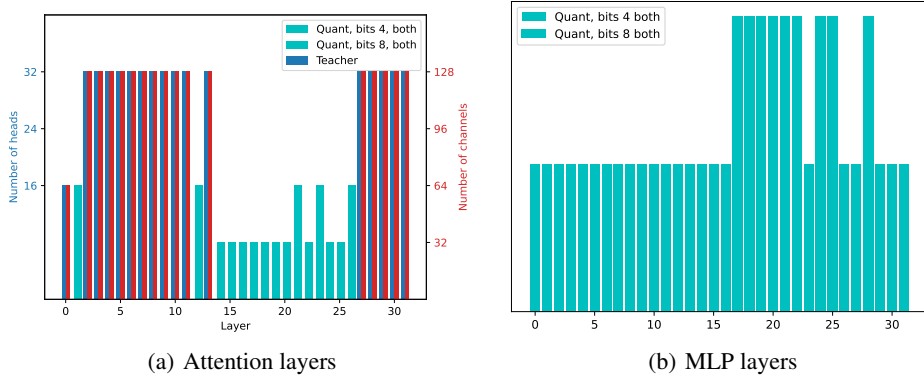

(a) Attention layers   (b) MLP layers

Figure 19: Layers parameters for a single architecture with parameters budget 50%. Includes quantized layers as candidates.

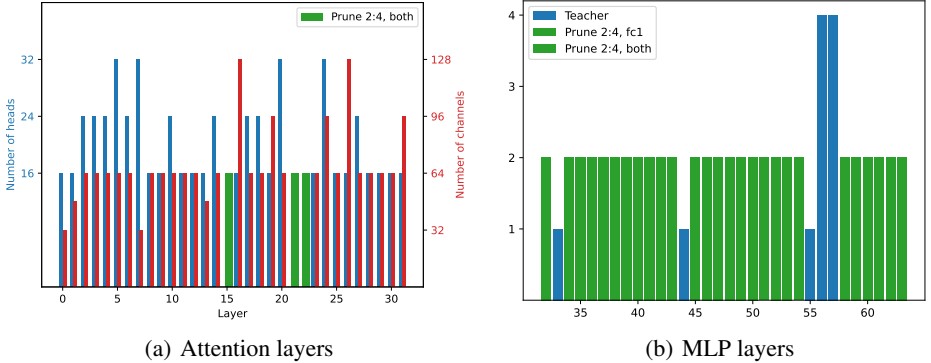

(a) Attention layers   (b) MLP layers

Figure 20: Layers parameters for a single architecture with parameters budget 50%, includes pruned layers as candidates.

