# OpenReview forum: "UNAST: Unified framework for Neural Architecture Search for Transformers"
_ICLR.cc/2025/Conference — ICLR 2025 Conference Withdrawn Submission_

### Official Review · Reviewer_kfsR · 2024-10-27

**Soundness:** 2
**Presentation:** 1
**Contribution:** 2
**Rating:** 3
**Confidence:** 4

**Summary:**

The paper introduces UNAST, a novel unified framework designed for post-training optimization of Large Language Models (LLMs). It integrates Neural Architecture Search (NAS) with sparsity and quantization techniques aimed at compressing LLMs. Starting from a pre-trained model, UNAST replaces inefficient layers such as attention and Multi-Layer Perceptron (MLP) with more efficient alternatives, adjusting parameters as well. The replacement layers undergo a process known as local distillation to replicate the performance of the original layers. The authors report that UNAST can reduce training costs significantly—up to 10 times compared to training smaller models from scratch—and improve latency and memory efficiency by up to 60% while maintaining high accuracy.

**Strengths:**

1.  By integrating NAS with sparsity and quantization, UNAST provides a holistic approach to compressing LLMs, including neural architecture search with sparsity and quantization technique.
2.  The method can lead to significant reductions in training costs and improvements in runtime efficiency and memory usage. UNAST  also allows for the creation of non-uniform models, providing insights into how different compression techniques affect Transformer layers.

**Weaknesses:**

1. The process involves several stages including layer-wise distillation and architecture search, which could make the methodology complex to implement.
2. The overall novelty of this idea is limited. It tries to use the previous compression techniques together and applies on the some language models. It is good as a technique report, while the current contribution is insufficient for a top-tier conference.
3. The organization of this paper is messy to some extent. For example, page 7 has a large blank space.

**Questions:**

1. What are the operational computing resources and energy consumption implications when deploying optimized models with UNAST compared to traditional methods?
2. How does the order of different compression techinques affect the final results? For example, we play NAS-spasitification-quantization, or spasitification-NAS-quantization, or perform these techinques simultaneously.

---

### Official Review · Reviewer_5Xyn · 2024-10-29

**Soundness:** 3
**Presentation:** 1
**Contribution:** 2
**Rating:** 5
**Confidence:** 3

**Summary:**

This paper introduces UNAST as a post-training optimization approach for LLMs. By applying NAS, UNAST reduces training costs by up to 10x compared to training a smaller model from scratch. Evaluations on LLaMA and GPT models demonstrate that UNAST improves latency and memory usage with minimal impact on accuracy.

**Strengths:**

1. This paper addresses the important challenge of efficient LLM deployment.
2. It takes a novel approach by applying NAS to LLMs, moving beyond traditional methods like quantization and sparsity.
3. The proposed method demonstrates improvements over baselines across various experimental settings.

**Weaknesses:**

1. The presentation could be enhanced for clarity. Are you suggesting that post-training is more efficient than training an LLM from scratch for deployment? However, if resources are available to train a large LLM, the cost of training a smaller model may not be prohibitive.

2. The authors should emphasize key results in both tables and text. For instance, highlighting the best-performing method in bold in Table 4 would be helpful, and it would be beneficial to specify the improvements in accuracy and latency achieved within the paragraphs.

**Questions:**

1. Could you clarify the source of GPT-3 in your experiments? To my knowledge, OpenAI has not released a GPT-3 checkpoint publicly.

---

### Official Review · Reviewer_PZYs · 2024-11-03

**Soundness:** 3
**Presentation:** 1
**Contribution:** 3
**Rating:** 5
**Confidence:** 4

**Summary:**

In this paper, the authors propose a neural architecture search (NAS) approach to large language models with affordable search computation costs and latency/num. of parameters constraints. In the methodology, the authors first adopt a layer-wise knowledge distillation, and then conduct architecture search / selection, and finally fine-tune the best model. Afterward, the authors conduct experiments to show the benefits of proposed UNAST, and showcase the findings in terms of efficient model design.

**Strengths:**

Efficient language model design is an important research topic in the community. In this paper, the authors make NAS algorithm affordable in the domain of large language models. Meanwhile, through a series of experiments, the authors provide several interesting findings, which could be guidance for efficient language model designs in the future.

**Weaknesses:**

1. The paper writing seems a little rush, which reduces the readability. Therefore are several notations and terms, and even conclusions, which are not explained or well interpreted. Please see the questions.

2. How the authors ensure the comparisons in Table 4 are fair considering there is no budget description of other approaches?

3. The title is a larger scope (transformer) while the paper mainly focuses on language model.

4. Finding 2 and finding 3 seem trivial to me: with less parameters, layer skipping is naturally worse than linear layer, and also worse than smaller alternatives. I think a performance/parameter tradeoff is necessary for visualization.

**Questions:**

I have many questions regarding the methodology and experimental results.

1. Line 147: what is the meaning of $j$ in $s_{ij}$? why there is a $i$ in  $X_i^D$.

2. Line 159: what is the meaning of $2N$? Does $N$ mean the number of transformer block?

3. Line 160, what is the meaning of $S^{M}$ and $S^A$? There should be $\\{0,1\\}^{S^M}$.

4. In equation 2, there should be a loss function for $(T(x), y)$.

5. Line 181: "if" $I$-th layer.

6. In equation 3, what is meaning of $\textbf{a}_i$, does it mean the accuracy?

7. In Line 264, what is the meaning of LM score? What kind of evaluation score?

8. in line 274, why only 15 model candidates are considered? From my understanding, in the local distillation, each layer could be substituted with C candidates. If there are 2N layers, the possible candidates should be 2CN. Please clarify how 15 is considered.

9. In Figure 3, what is the meaning of ns? Does it mean the intermediate size of FFN?

10. In Line 377, how to conclude that "The full pool of candidates shows the best overall result" considering "full pool" is suboptimal compared to other designs in LM-eval. It only has advantages in Wiki ppl.

11. In Table 4, please use uparrow or downarrow to show the performance gains or losses.

---

### Official Review · Reviewer_9Trh · 2024-11-03

**Soundness:** 3
**Presentation:** 1
**Contribution:** 2
**Rating:** 3
**Confidence:** 3

**Summary:**

In this paper, the authors propose a neural architecture search method, UNAST, to compress LLMs post training based on the desired model size and latency.

Specifically, UNAST:
1. pre-define the search space on the attention and MLP block, like the number of heads and KV dimention of the attention block, and projection dimention of the MLP block and so on.
2. applies layer-wise knowledge distillation from the original later (techer layer) to the modified layer (student layer), and obtains the scores for various candidates.
3. selects the highest ranked candidates based on the cumulative accuracy metric, and finetunes the modified models with limited number of tokens.

Extensive experiemnts show some instructive findings:
1. Modifying the attention block is easier than modifying the MLP block;
2. Reduing the number of heads is easier than reducing the KV dimentions;
3. Pruning the first-half layers is easier than the second-half.

The final pruned LLMs are smaller and faster, and also outperforms or is comparable to various baselines.

**Strengths:**

1. The experiments are extensive, with most claims well supported by experiments.
2. The final LLM shows the effectiveness of UNAST, with smaller model size and higher latency, but still comparable to most baselines.

**Weaknesses:**

1. **Poor presentation**: The paper is not easy to follow because of the following problems:

(1) The citation format: Like in L265, the citation looks like "the zero-shot lm-harness benchmark Gao et al. (2023) accuracies". For such citation, you should use \citep{} instead of \citet{}. I have obsered quite a few such mistake.

(2) The legend in the figures are not well-explained. Taking Figure 4 as an example, I can guess "att_nh_32_kv_96" means 32 and 96 refer to the number of attention heads and KV dimentions. But for "mlp_hs_1_swiglu_1", I don't really understand the meaning of 1. I suppose it means the dimention factor in the MLP. I believe a well-explained notation is necessary for them.

(3) The experimental section is too messy, I suggest a better structure of results. You should leave more space and explanation to the important results or findings.

(4) Better to highlight some results: In Table 4, it's better to highlight the best results. Otherwise, it's not easy to observe the the improvement of UNAST.

2. **Lack of novelty**: Please correct me if I understand wrongly. UNAST applies NAS to pre-trained LLMs. Compared to previous works that apply NAS to find an optimal architecture and train it from scratch, UNAST tries to find a compact LLM based on a pre-trained LLM. I think such a novelty is not enough for ICLR.

**Questions:**

Please see some questions in weakness.

---

### Note · Authors · 2024-11-25

I have read and agree with the venue's withdrawal policy on behalf of myself and my co-authors.